# Rheumatoid Arthritis and CLOVES Syndrome: A Tricky Diagnosis

**DOI:** 10.3390/diagnostics10070467

**Published:** 2020-07-09

**Authors:** Laura Damian, Andrei Lebovici, Cristina Pamfil, Cristina Belizna, Romana Vulturar

**Affiliations:** 1Cab. Reumatologie Dr. Damian, 6-8 Petru Maior St, 400002 Cluj-Napoca, Romania; ldamian.reumatologie@gmail.com; 2Rheumatology Department, Emergency Clinical County Hospital Cluj, Center for Rare Musculoskeletal Autoimmune and Autoinflammatory Diseases, 2–4 Clinicilor St, 400006 Cluj-Napoca, Romania; 3Radiology Department, “Iuliu Hațieganu” University of Medicine and Pharmacy Cluj, 8 Victor Babeș St, 400023 Cluj-Napoca, Romania; 4Center for Rare Musculoskeletal Autoimmune and Autoinflammatory Diseases, “Iuliu Hațieganu” University of Medicine and Pharmacy Cluj, 8 Victor Babeș St, 400023 Cluj-Napoca, Romania; cristinapamfil.umfcluj@gmail.com; 5Vascular and Coagulation Department, University Hospital Angers, 49100 Angers, France; cristina.belizna@wanadoo.fr; 6UMR CNRS 6015, INSERM U1083, Rue Haute de Reculée, 40945 Angers CEDEX 01, France; 7Department of Molecular Sciences, “Iuliu Hațieganu” University of Medicine and Pharmacy Cluj, 6, Pasteur St., 400349 Cluj-Napoca, Romania; romanavulturar@yahoo.co.uk; 8Cognitive Neuroscience Laboratory, University Babeș-Bolyai, 30, Fantanele St., 400294 Cluj-Napoca, Romania

**Keywords:** CLOVES syndrome, rheumatoid arthritis, macrodactyly, PIK3/AKT/mTOR pathway, *PIK3CA*-related overgrowth spectrum

## Abstract

The PI3K/AKT/mTOR signaling pathway is significantly activated in rheumatoid arthritis. In addition, somatic activating mutations of the PI3K/AKT/mTOR pathway may result in *PIK3CA*-related overgrowth spectrum diseases, including CLOVES (Congenital Lipomatous Overgrowth, Vascular malformation, Epidermal nevi, Skeletal abnormalities/Scoliosis) syndrome. We describe the case of a young female patient, with anti-citrullinated peptide antibodies-positive rheumatoid arthritis, referred for persistent finger pain and stiffness. Examination revealed discrete macrodactyly involving two fingers, scoliosis, asymmetrical calves, venectasias, a shoulder nevus and triangular feet with a “sandal gap” between two toes. These mild dysmorphic features with early-onset and the history of surgeries for thoracic lipoma and venous malformation were strongly suggestive of CLOVES syndrome. Confirmatory mutation analysis was not performed, as blood or saliva testing is not contributive for tissue-specific localized effects in the *PIK3CA*-related overgrowth spectrum. Nevertheless, lack of detection of a *PIK3CA* mutation does not exclude the diagnosis in patients fulfilling clinical criteria. Due to the patient’s wish to plan a pregnancy, therapy consisted in sulfasalazine and hydroxychloroquine, along with orthotic correction of leg length discrepancy. Overgrowth syndromes and arthritis may share common pathways. Mild macrodactyly should be differentiated from dactylitis. Diagnosing patients with minimal dysmorphic features within the PI3K-related overgrowth spectrum may help design better care strategies, in the quest for personalized medicine.

## 1. Introduction

PI3K/AKT/mTOR is a cell cycle signaling pathway regulating cell growth, metabolism, proliferation, survival and migration [1]. The PI3K (phosphatidylinositol 3, 4, 5 triphosphate kinase) family regulates many important cellular mechanisms. Upon ligand binding, PI3K activates AKT (or protein-kinase B), an apoptotic molecule inhibitor, and activates downstream mTOR (mammalian target of rapamycin), involved in cell growth and proliferation [1].

PI3K/AKT/mTOR activation is important in oncogenesis; somatic activating PI3K mutations during embryogenesis lead to the PIK3CA-related overgrowth spectrum (PROS) [2]. Dysregulated PI3K/AKT/mTOR in immune cells is found in autoimmune diseases, including rheumatoid arthritis (RA) [1]. PI3K isoforms are key enzymes in leukocyte signaling in RA [1,3].

## 2. Case

A young female patient, non-smoker, presented for small joint pain mainly of the hands. She had been diagnosed two years prior with anti-citrullinated peptide antibodies (ACPA)-positive RA and treated with sulfasalazine with partial improvement. Methotrexate was refused by the patient, wishing to plan a pregnancy. Her history included surgery for a left anterior thoracic lipoma, and three years prior a left calf varicose vein stripping, which revealed a venous malformation (small saphenous vein much larger than the great saphenous vein). She had no history of sacroiliitis and no personal or familial history of psoriasis, inflammatory bowel disease or uveitis. No rheumatoid arthritis, spondylarthritis or diseases evolving with dysmorphic features were present in the family. Her first pregnancy was uneventful.

Physical examination revealed discrete enlargement of the 2nd and 3rd right fingers (Figure 1c) with limited mobility, reported by the patient as being larger ever since she remembered it, and tender interphalangeal joints. In addition, she presented scoliosis and a longer and enlarged left calf (one centimeter in length and 4 cm in circumference compared to the right calf), larger triangular feet with a “sandal gap” between the 3rd and the 4th right toes, venous ectasias mainly on the left lower limb, and a left shoulder nevus (Figure 1).

The laboratory findings were notable for mild inflammation [erythrocyte sedimentation rate (ESR) of 28 mm/h (normal <12 mm/h), and C-reactive protein (CRP) of 1.2 mg/dL (normal <0.6 mg/dL)], elevated ACPA (88 IU/mL, normal <20 IU/mL) and high rheumatoid factor (128 IU/mL, normal <8 IU/mL). Radiographies showed juxta-articular osteoporosis, enlargement of the 2nd and 3rd first finger phalanges of the right hand and a larger distance between the 3rd and 4th right toes (Figure 2). Right hand MRI revealed discrete bone enlargement in the involved fingers (Figure 2), as well as joint effusions (Appendix A). Abdominal ultrasonography was normal. An enhanced computed tomography of the thorax, abdomen and pelvis did not find any vascular or lymphatic malformation.

This clinical picture with congenital onset was diagnostic for a mild CLOVES (Congenital Lipomatous Overgrowth, Vascular malformation, Epidermal nevi, Skeletal abnormalities/Scoliosis) syndrome belonging to PROS, associated with RA. Genetic testing was considered, but not performed, as blood or saliva DNA testing is non-contributive in PROS for tissue-specific, localized effects [2]; the biopsies of the removed lipoma and vascular tissue were not available.

Given the patient’s wish for a second pregnancy, sulfasalazine was increased to 3 g/day, and hydroxychloroquine (400 mg/day) was added, along with orthotic correction of the leg length discrepancy and kinetotherapy. After 3 months, the disease activity was low (DAS28 score 3.01, ESR 13 mm/h, CRP 0.6 mg/dL).

## 3. Discussion

As RA is a chronic autoimmune disease that affects the joints generally in a symmetric manner, the discretely enlarged fingers in our patient raised the initial suspicion of spondylarthritis or psoriatic arthritis; instead, the macrodactyly was a mild manifestation of CLOVES syndrome.

CLOVES syndrome (Congenital Lipomatous Overgrowth, Vascular malformation, Epidermal nevi, Skeletal abnormalities/Scoliosis) is a localized overgrowth syndrome, due to mutations of PI3K gene [2]. CLOVES diagnosis relies on congenital or early childhood onset of overgrowth, with sporadic and patchy or irregular distribution, in the presence of the somatic *PIK3CA* mutation, which can be demonstrated only on biopsies from the affected tissues, where mosaicism may also be present [2]. Lack of detection of a *PIK3CA* mutation classifies the disease as probable, but does not exclude the diagnosis in patients fulfilling clinical criteria [2]. Two or more features of the following are required: overgrowth (adipose, muscle, nerve, skeletal), vascular malformation, epidermal nevi, or isolated features such as isolated macrodactyly, truncal adipose overgrowth, hemimegalencephaly or focal cortical dysplasia, seborrheic keratosis or benign lichenoid keratosis [2]. Our patient clinically fulfilled the criteria, even if the features were mild, having an overgrowth of two digits and of the left calf, a ”sandal gap”, scoliosis, a removed thoracic lipoma, vascular malformation and an epidermal nevus.

As the PI3K/Akt/mTOR pathway regulates cell proliferation and migration, its role in RA synovial hypertrophy and immune cell infiltration now being extensively explored. The PI3K/AKT/mTOR signaling pathway is activated in various cell lines in RA, such as macrophages, fibroblast-like synovial and endothelial cells, and the expression of the negative regulatory molecule PTEN is reduced [1,4,5]. Activation of mTOR links synovitis and structural damage in RA [4,6], but also links inflammation and metabolic dysregulation [3]. (Appendix A). 

Arthritis was reported in other diseases evolving with mTOR hyperactivation, such as Cowden’s syndrome due to PTEN deficiency [7], and in the recently described activated PIK3kδ syndrome or APDS, with combined immune deficiency and often autoimmune manifestations [8].

Upstream regulators of the PI3K/AKT/mTOR signaling pathway are being developed to treat autoimmune, proliferative and degenerative diseases [4]. BYL719, a targeted inhibitor of PIK3CA, improved organ dysfunction in nineteen patients with PROS, harboring complex vascular malformations [9]. Rapamycin (sirolimus) alleviated juvenile idiopathic arthritis in a kidney transplant recipient [10]. Sulfasalazine inhibits the AKT pathway and promotes autophagy [11]. Among the common medications, COX-2 inhibitors decrease cell proliferation and induce RA-FLS apoptosis by reducing AKT signaling [12].

In our patient, sulfasalazine was continued, along with hydroxychloroquine, in the perspective of a pregnancy. The somatic (as opposed to germline) mutation markedly decreases the risk of transmitting the CLOVES syndrome to the offspring. The follow-up in CLOVES patients also includes the detection of central phlebectasia to decrease the risk of pulmonary embolism [2]. Although the malignancy risk is unknown and probably low, tumoral screening is recommended [2], and is even more relevant when biologic anti-TNF therapy is contemplated for RA.

Overgrowth syndromes and arthritis may share common pathways. Mild macrodactyly should be differentiated from dactylitis. Diagnosing patients with minimal dysmorphic features within the PI3K-related overgrowth spectrum may help design better care strategies, in the quest for personalized medicine.

The patient has given informed consent for publishing the case report.

## Figures and Tables

**Figure 1 diagnostics-10-00467-f001:**
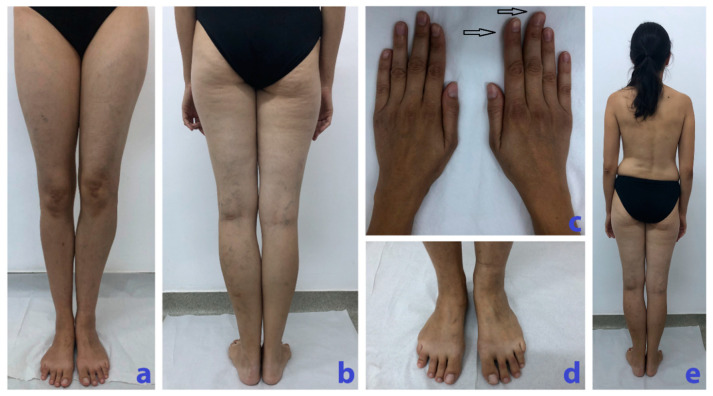
(**a**) Longer left calf with slight deformity. (**b**) Longer and larger left calf with vascular ectasias. (**c**) Hands with discrete enlargement and ax deviation of the 2nd and 3rd right fingers with macrodactyly (arrows). (**d**) Larger triangular right foot with a “sandal gap” between the 3rd and the 4th toes. (**e**) Scoliosis, longer and larger left calf.

**Figure 2 diagnostics-10-00467-f002:**
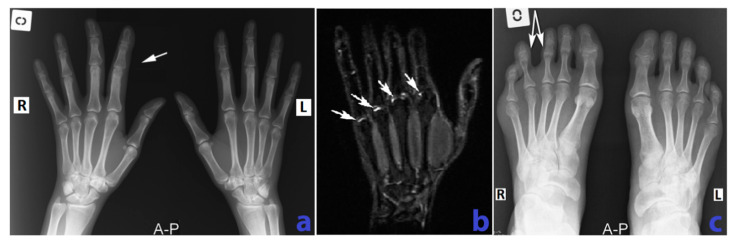
(**a**) Comparative x-ray of the hands, AP (anteroposterior) view shows mild “band like” osteoporosis adjacent to the MCP (metacarpophalangeal) joints, bilateral and slightly enlarged 2nd and 3rd finger on the right hand associated with slight ulnar deviation of the third finger. (**b**) MRI of the right hand, STIR (Short Tau Inversion Recovery) sequence, coronal view demonstrates mild fluid collections in the metacarpophalangeal joint spaces (white arrows). (**c**) Comparative x-ray of the feet AP view shows enlargement interdigital space with “sandal gap” appearance between the 3rd and the 4th right toes (white arrows). R—right, L—left.

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
