# Peer review of "Rheumatoid Arthritis and CLOVES Syndrome: A Tricky Diagnosis"

_diagnostics, 2020, doi:10.3390/diagnostics10070467_

Round 1

Reviewer 1 Report

Dear Authors,

In my opinion this case report is well written and interesting. This manuscript has also clinical importance. However before the acceptance I suggest to add the permission of patient as a supplementary meterial. There is only one short sentence that: "The patient has given informed consent for publishing the case report."

Author Response

Dear Authors,

In my opinion this case report is well written and interesting. This manuscript has also clinical importance. However before the acceptance I suggest to add the permission of patient as a supplementary meterial. There is only one short sentence that: "The patient has given informed consent for publishing the case report."

Dear Editor,

Thank you, we appreciate the comments made by the reviewers; here is the manuscript with the corrections that are made using track changes:

At the line 29: Due to the patient’s wish

Kind regards,

Dr. Andrei Lebovici

Reviewer 2 Report

The paper by Damian et al describes a case of CLOVES syndrome. The case is well described and indeed completely fits in the clinical picture.

This case report cannot – in my opinion – be published without citing Venot et al. Nature 2018;558:540-6; these authors treated PROS patients with a PIK3CA inhibitor successfully.

For the rest, it’s a nice paper that can be published.

Minor:
Line 72: “first” fingers? Should this be right or left?
Line 92: Please remove the S from DISCUSSION

Author Response

Dear Editor,

Thank you, we appreciate the comments made by the reviewers; here is the manuscript with the corrections that are made using track changes:

  1. At the lines 72-73 was introduced: "at the right hand".
  2. At the line 92: the word "Discussions" was replaced by "Discussion".
  3. At the lines 119-121: a phrase regarding the targeted therapy, according to Venot et al., 2018.
  4. At the lines 122-124: due to the introduction of ref. Venot et al, the old references 9, 10, and 11 became 10, 11, 12.

Additionally, there are two corrections:

At the line 29: Due to the patient’s wish

At the line 106: having an overgrowth

Kind regards,

Dr. Andrei Lebovici